# Reformulation of Trivers–Willard hypothesis for parental investment

Jibeom Choi[1✉], Hyungmin Roh [2], Sang-im Lee [3✉], Hee-Dae Kwon [4✉], Myungjoo Kang [5✉] & Piotr G. Jablonski [6,7✉]

The Trivers-Willard hypothesis (TWH) plays a central role in understanding the optimal investment strategies to male and female offspring. Empirical studies of TWH, however, yielded conflicting results. Here, we present models to predict optimal comprehensive multi-element parental strategies composed of primary sex ratio, brood size, resource allocation among offspring, and the resultant secondary sex ratio. Our results reveal that the optimal strategy depends on sex differences in the shape of offspring fitness function rather than in fitness variance. Also, the slope of the tangent line (through the origin) to the offspring fitness function can be used to predict the preferred offspring sex. We also briefly discuss links between the model and the empirical research. This comprehensive reformulation of TWH will offer a thorough understanding of multi-element parental investment strategies beyond the classical TWH.

[1] Department of Applied Mathematics, Kyung Hee University, Yongin, Republic of Korea. [2] Samsung Research, Samsung Electronics Co., Ltd., Seoul, Republic of Korea. [3] Laboratory of Integrative Animal Ecology, Department of New Biology, DGIST, Daegu, Republic of Korea. [4] Department of Mathematics, Inha University, Incheon, Republic of Korea. [5] Department of Mathematical Sciences, Seoul National University, Seoul, Republic of Korea. [6] Laboratory of Behavioral Ecology and Evolution, School of Biological Science, Seoul National University, Seoul, Republic of Korea. [7] Behavioral Ecology Group, Museum and Institute of Zoology, Polish Academy of Sciences, Warsaw, Poland. ✉email: snu10@snu.ac.kr; sangim@dgist.ac.kr; hdkwon@inha.ac.kr; mkang@snu.ac.kr; snulbee@behecolpiotrsangim.org

In their original verbal argument, Trivers and Willard[1] predicted that the offspring sex ratio at independence (i.e., the secondary sex ratio) should be optimally adjusted depending on maternal condition. Trivers and Willard[1] assumed that good maternal condition results in higher maternal investment and that the fitness (analogous to offspring's reproductive value[2]) accrued by parents through their offspring depends on parental investment that is inequitable among offspring. In polygynous mammals, the fitness of a female offspring often shows low variance/range, defined as the difference between the maximal and minimal values of the fitness, due to the limitations imposed by gestation and nurturing. On the other hand, the male offspring fitness exhibits higher variance, because only a few superior males succeed in the competition for mating to the majority of females, while the remaining males are excluded from such mating. Thus, mothers in good condition, who can produce superior male offspring, are expected to produce male-biased broods. In the same line of reasoning, the optimal decision for mothers in poor condition, who are unable to invest sufficiently to produce superior males, is to produce female-biased broods to gain the greatest possible fitness returns. Thus, the general prediction from the classical TWH is concerning the allocation of investment into male and female offspring in a brood determined by the variance in the fitness that can be accrued from an offspring. Although originally proposed for polygynous mammals, Trivers–Willard hypothesis (TWH) can be applied to numerous species with sex differences in fitness accrued by parents from offspring. These differences may be caused by various factors such as sex-specific costs of rearing, sex-specific growth curves, sexual size dimorphism, local mate competition, and local resource competition[3,4]. While the theory has brought about numerous empirical studies about certain elements of parental investment (including primary sex ratio[5,6] and sex-biased provisioning to offspring[7,8]), the studies have produced conflicting empirical evidence[9–11]. To elucidate the principles underlying the contrasting parental strategies, we have built comprehensive mathematical and computational models.

The models described in this paper demonstrate that the sex difference in the shape of fitness functions is central for understanding the variation in the empirical evidence that is for or against TWH[12]. The concept of the fitness function, defined as the relationship between investment toward an offspring ($x_i$) and the fitness returns that parents accrue through this offspring, has already been used in other models[2,12–18], with the recent focus on population effects into the TWH[12,16]. However, the existing models to our knowledge do not simultaneously consider all elements of parental strategy: primary and secondary sex ratio (1° SR, 2° SR), clutch size, and the rules of resource allocation among offspring[19] (Fig. 1a; definitions are given in the Methods section).

Here, we explore how sex differences in the logistic (sigmoid) fitness functions[13,17,20–22] affect the multiple elements of the optimal parental strategy. We created various shapes of sigmoid fitness function by changing the parameters of the logistic equations (Fig. S1, Table S1) representing fitness accrued by parents from one offspring ($f_{b_i}(x_i)$) as a function of parental *per capita* investment ($x_i$). Although the shape of fitness functions in nature depends on multiple factors, we simplified the situation by postulating that condition-dependent effects of parents on offspring are mediated entirely through the amount of expendable parental investment (henceforth, $S$). We assumed that $S$ can be an approximate representation of parental condition. In reality, $S$ would depend on (1) the resources available to parents prior to and during a breeding event and (2) the abilities of parents (behavioral, physiological, etc.) to extract and transform those resources into the expendable

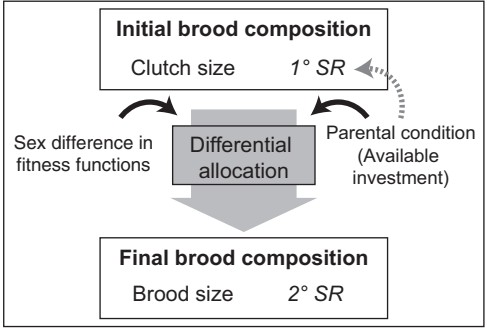

**Fig. 1 Schematics illustrating links between the components of the multi-element parental strategy.** Primary sex ratio (1° SR), clutch size, brood size, parental investment rules of differential allocation between sexes, and the resulting secondary sex ratio (2° SR), including two factors (parental condition/available investment and sex difference in fitness functions) that affect the optimal parental strategy are shown. The globally optimal strategy assumes the ability of parents to modify the primary sex ratio according to the parental investment, while the locally optimal strategy does not.

parental investment, both of which can be regarded as the components or determinants of parental condition. It was supposed in the model that, for a given $S$, parents should expend all of $S$ without consideration of upcoming breeding seasons. Therefore, our model is most appropriate for semelparous organisms. In addition, for each of the given $S$, the optimization process is performed under the condition that $S$ is constant during a reproductive event. In other words, there is no temporal variation of $S$ during reproduction and rearing. Therefore, the optimal relationship between $S$ and the parental strategy refers to the full set of optimal strategies for each of fixed $S$. Other assumptions, including over-investment, are presented in Supplementary Materials (henceforth, SM), Part 1. Within this framework, we aimed to determine how multiple elements of the optimal parental strategy respond to changes of $S$, and whether model predictions conform to the classical TWH or not, which accommodates the possibility of the reversed TWH predictions[11,12].

## Results and discussion
We considered a situation where the clutch is initially (i.e., at the beginning of parental investment) composed of $N = 10$ offspring. Offspring fitness functions are either the same for every individual (Fig. 2a; Fig. S1a, e) or different between the two types of offspring (sexes; Figs. 3a, 4a; Figs. S3–S16). In our model, parents can invest differentially in each offspring. Using analytical and computational modeling, we determined the multi-element parental strategy that maximizes the fitness accrued by parents (which is proportional to the sum of offspring fitness) from the whole brood.

**Broods with fitness functions identical for males and females.** We first looked at the situation when the fitness functions were identical for all the offspring (e.g., Fig. 2a). We figured out that choosing some of the offspring (Fig. 2b) and providing equitable *per capita* investment, $x_i$, (Fig. 2c) to each of them while abandoning the others is optimal, which is proven mathematically as well (Theorem 1 and its corollary in SM, Part 4). The result shows that the number of offspring that receive the investment increases monotonically with $S$. The monotonic increase can be represented by the summation of step functions against $S$ (Theorem 2 in SM, Part 4). Therefore, maximal clutch size (10 in this model) would only affect the upper bound of the number of offspring receiving investment. In addition, these results comply with the classical theories of the optimal clutch/brood size[23,24].

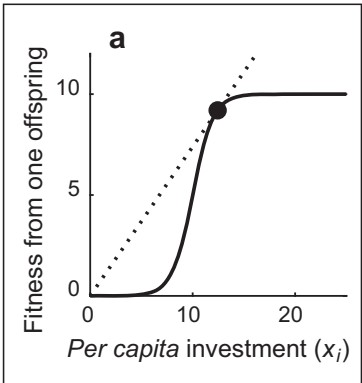
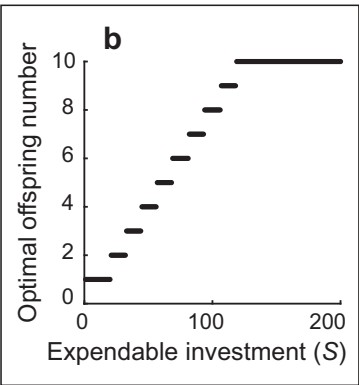
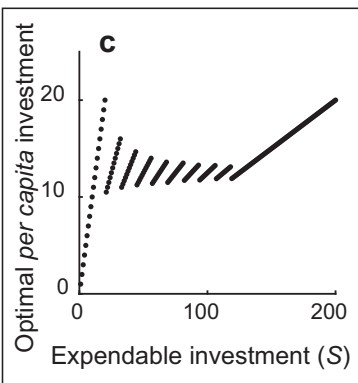

**Fig. 2 Results of the model *M1* (fully presented in Fig. S1a–d) where all offspring fitness functions are identical. a** The fitness function in model *M1*: the relationship between *per capita* investment in one offspring ($x_i$) and the fitness of this offspring. The most efficient point, where $f(x)/x$ is the greatest, is marked with a black circle. **b** The relationship between the amount of expendable investment (*S*) and the optimal number of offspring (optimal brood size) receiving care from parents that maximize the total brood fitness. **c** The relationship between the amount of expendable investment (*S*) and *per capita* investment ($x_i$) for optimal brood size (shown in **b**). The results of **b**, **c** are confirmed by mathematical analysis.

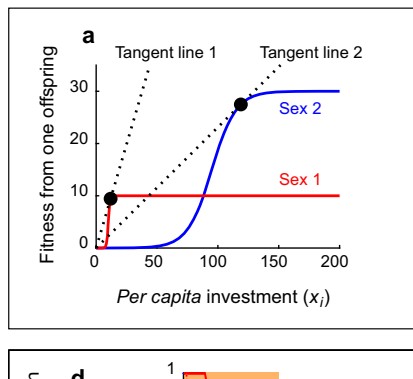
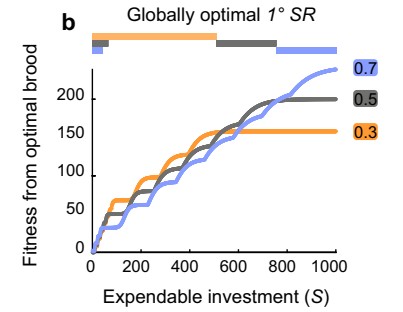
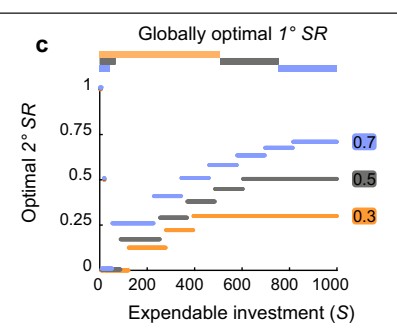
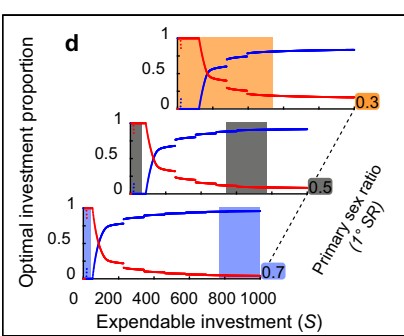
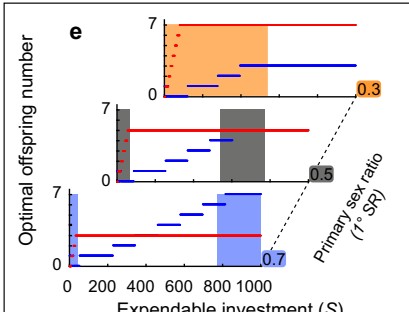
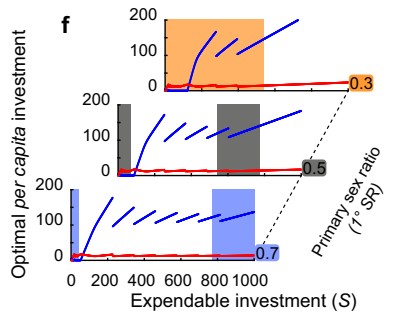

**Fig. 3 The results from model *M3* (fully presented in Fig. S3), in which predictions from the classical TWH hold with respect to the total number of offspring receiving care and total investment toward the specific sex: *Sex 2* is preferred over *Sex 1* when *S* is high. a** The fitness functions of *Sex 1* (red line) and *Sex 2* offspring (blue line) in model *M3*. In this model, the tangent line of *Sex 1* is steeper than that of *Sex 2*. **b** The relationship between total investment (*S*) and fitness accrued from the whole brood for three different primary sex ratios (0.3, 0.5, 0.7; yellow, gray, and light blue, respectively) by parents that optimize their parental strategy. The horizontal color bars on top of the panel represent the ranges of *S* in which the specific primary sex ratio(s) yield the highest value of fitness accrued from the optimal brood. The same bars are presented in **c** and these regions of *S* are indicated as shaded regions of the same color in **d**–**f**. **c** The relationship between total investment (*S*) and the brood's secondary sex ratio for the three different primary sex ratios (same color schematics as in **b**) for parents that optimize their investment. **d–f** The relationship between total investment (*S*) and the three aspects of the optimal parental strategy: proportion of investment (**d**), optimal numbers of offspring chosen for investment (**e**), and *per capita* investment ($x_i$) into *Sex 1* (red) and *Sex 2* (blue) offspring (**f**) for three different primary sex ratios. Color-shaded (same color schematics as in **b**) regions indicate the globally optimal strategy: in each primary sex ratio, shadings indicate the range of *S* in which fitness from the brood is globally maximized (as seen in **b**).

**Broods with fitness functions different for males and females**. Then, we focused on broods composed of two sexes, *Sex 1* and *Sex 2* (marked in red and blue lines, respectively, in Figs. 3, 4a), that differed in the shape of the fitness functions. Figures 3 and 4 represent two examples (models *M3* and *M5*) chosen from all computational models that simulate broods with 14 cases of different fitness function pairs (SM, Part 3). In all pairs of *Sex 1* and *Sex 2* functions, it holds that *Sex 1* has greater fitness when *per capita*

investment is small (see Methods). For each pair of fitness functions, we simulated three types of clutches with different *1° SRs*: *Sex-1*-biased (*1° SR* = 0.3; for convenience, we express sex ratio as the proportion of *Sex 2* offspring in the clutch), equal (*1° SR* = 0.5), or *Sex-2*-biased (*1° SR* = 0.7) clutches. Three sex ratios were marked with orange, gray, and light blue in Figs. 3, 4, respectively.

For each pair of fitness functions (Figs. 3a, 4a; panels (a) in Figs. S3–S16), we calculated the fitness accrued by optimally

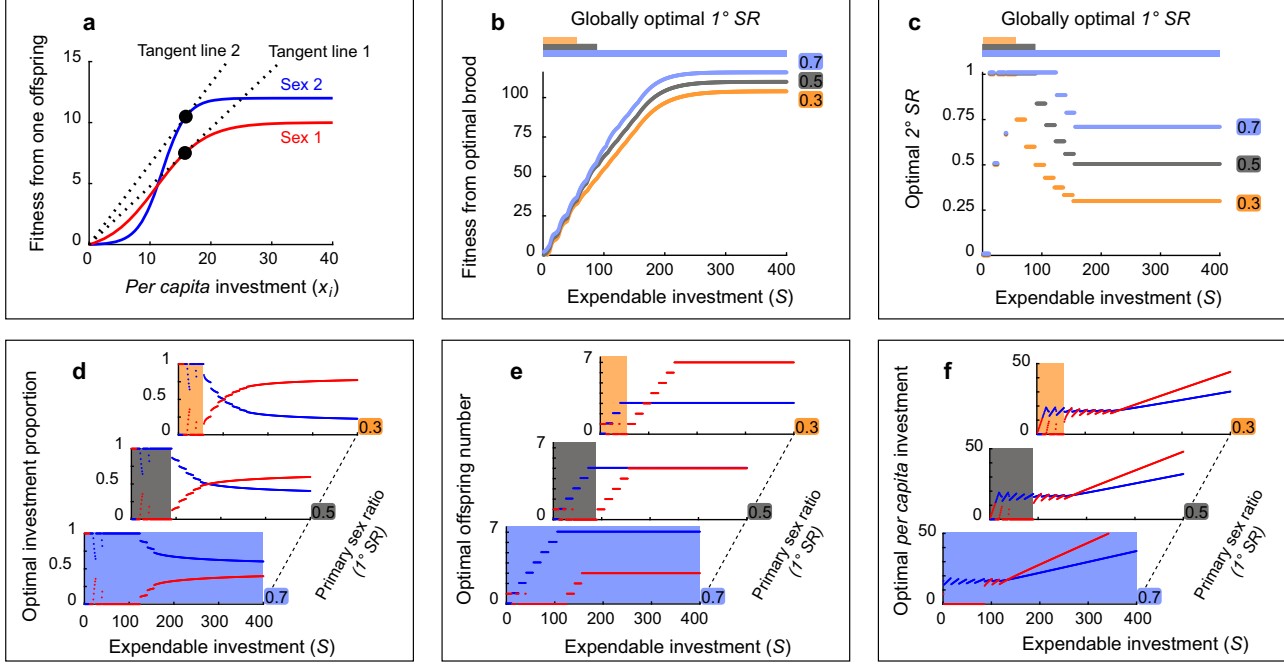

**Fig. 4 The results from model *M5* (fully presented in Fig. S5), in which the tangent line of *Sex 2* is steeper than that of *Sex 1*. The predictions are inconsistent with the classical TWH.** Panels represent the same information as in panels from Fig. 3. In this model, the reversed TWH predictions are observed: *Sex 1* offspring (red lines) receive most investment at high parental expendable investment (*S*) for equal or *Sex-1*-biased primary sex ratios, despite having a lower variance in fitness.

allocating parents from a brood of a given *1° SR* as a function of *S*. This resulted in three functions representing the three *1° SRs* (Figs. 3b, 4b). Then, we determined ranges of *S* in which a specific *1° SR*, or a combination *1° SRs*, is optimal (i.e., greatest total offspring fitness). The details of the globally optimal parental strategies are marked by color-shaded rectangles in Figs. 3d–f, 4d–f (details for all models are in Figs. S19-S32 and Table S3). The globally optimal strategy is composed of the effects of *S* on the combination of four elements: *1° SR* (Figs. 3b, 4b), *2° SR* (Figs. 3c, 4c), investment allocation rules (Figs. 3d, f, 4d, f), and brood size (The number of males and females in Figs. 3e, 4e). The results also allowed us to analyze the locally optimal (i.e., within each *1° SR*) parental investment strategies (Table S2) simulating organisms that do not have control over *1° SR* but can control their investment allocation rules and the resultant *2° SR*.

We analyzed if and how the difference in fitness function shapes between *Sex 1* and *Sex 2* affects the locally and globally optimal relationships. The fitness function of *Sex 1* in models *M3* and *M5* assumes that the fitness increases rapidly with a relatively small investment ($x_i$), but the maximum value is small compared to that of *Sex 2*. The fitness function of *Sex 2* assumes a slower initial increase in fitness against $x_i$, but the maximum fitness is larger than that of *Sex 1*. In many species, these fitness functions of *Sex 1* and *2* correspond to those of female and male offspring, respectively[14].

Consider the case where the two fitness functions differ as in Fig. 3a (model *M3*). In this situation, all the elements of the globally optimal parental strategy that maximizes fitness vary in accordance with the classical TWH. That is, the sex of offspring with a higher variance in fitness (i.e., *Sex 2*) is preferred by the parents who have the greater amount of *S*. This also holds true for locally optimal strategies within each *1° SR* (Fig. 3d–f). For each *1° SR*, the three aspects of optimal allocation strategy (Fig. 3d–f) change with *S* in accordance with the classical TWH: *Sex-1*-biased care for the parents with small *S* and *Sex-2*-biased care for those with large *S* as evidenced in Fig. 3d. Similar results were generally

observed for several other models (Tables S2, S3). Notice the apparent scattering of data points or non-continuity of lines across the models (Figs. 3d, 4d), which appears to be associated with the changes in the optimal brood composition along *S* (Figs. 3e, 4e).

The optimal *1° SR* changed in accordance with the TWH predictions for all models as *S* increased. While the brood's fitness did not depend on *1° SR* for small *S* in model *M5* (from 0 to 55; overlapping lines in Fig. 4b), the optimal *1° SR* was *Sex-2*-biased for the larger values of *S* (>89; the light blue line is higher than the others on this domain in Fig. 4b), which is similar to the classical TWH predictions. Consequently, in situations similar to Fig. 4a, the natural selection may favor the ability of the parent(s) to optimally adjust the *1° SR* in a manner resembling the classical TWH predictions (male-biased *1° SR* is preferred for larger *S*).

However, the locally optimal parental investment rules in model *M5* (Fig. 4d–f), as well as some other models (Figs. S7, S8, S11–S16; Table S2), were inconsistent with the classical TWH, exhibiting trends (at least for some of the *1° SRs*) contrary to the classical TWH. For each of the three *1° SR* values considered in model *M5*, it is optimal to invest more toward *Sex 2* offspring (Fig. 4d) and to rear more of them when *S* is small though *Sex 2* has a higher variance in fitness (Fig. 4a). In these cases, the resultant *2° SR* also varied in a manner inconsistent with the classical TWH that predicts increasing *2° SR* with increasing *S*.

To explore the effects of the parents' allocation capability on the evolution of *1° SR*, we compared the fitness from optimal allocation strategies with those from equitable distribution of investment among all offspring for each *1° SR* (dashed lines in panels (b) of Figs. S19–S32). Suppose that the maximum fitness of *Sex 2* is greater than that of *Sex 1*. In these models with equitable distribution, optimal *1° SR* follows the pattern reminiscent of classical TWH: *Sex-1*-biased *1° SR* is optimal when *S* is low, and *Sex-2*-biased *1° SR* is optimal when *S* is high. This pattern is mathematically supported in Theorem 6 (SM, Part 4). Hence, even if parents are unable to optimize the investment allocation

strategies (i.e., allocation is outright equitable to all offspring), the optimal *1° SR* changes with *S* in accordance with the classical TWH. However, there are differences in switch points of optimal *1° SR* for optimal and equitable strategies, indicating selection pressures for primary sex ratio adjustment and optimal parental allocation.

As pointed out by Hewison and Gaillard[11], and Veller et al.[14], the meaning of the classical phrase that parents prefer one sex over the other is ambiguous in the original TWH[1]. It may be interpreted as bias in any elements of the parental strategy considered in our comprehensive model. Our model allows us to determine biases in every element of parental investment and how the biases vary depending on *S*. For example, according to the globally optimal strategy of model M3, a switch from *Sex-1*-biased *1° SR* (= 0.3) to equal *1° SR* (= 0.5) occurs at *S* = 540, and a switch from equal to *Sex-2*-biased *1° SR* (= 0.7) occurs at *S* = 770 (Fig. 3b). On the other hand, the switch from generally investing more *S* into whole *Sex 1* offspring in a brood to investing more *S* into whole *Sex 2* offspring in a brood occurs at *S* = 170 when *1° SR* = 0.3 (note the intersection of two lines in the upper graph of Fig. 3d).

Similar differences in the location of the switching points for different elements of the parental strategy can be observed within locally optimal strategies. For example, a change from *Sex-1*-biased optimal *2° SR* (<0.5) to *Sex-2*-biased optimal *2° SR* (>0.5) occurs over the range of *S* between 344 and 460 given that *1° SR* = 0.7 (Note the range of *S* on which the *y*-value of the light blue line is 0.5 in Fig. 3c). This is the same as the range where red and blue lines overlap in Fig. 3e for *1° SR* = 0.7. On the other hand, within the locally optimal strategy for *1° SR* = 0.7, we observed a switch from *Sex-2*-biased to *Sex-1*-biased investment (with respect to the total investment to each sex) at *S* = 88 (Fig. 3d). Thus, our model not only provides a comprehensive quantitative view on the sex ratio version and the investment version of the TWH[14] but also allows us to estimate switches of sex-biased investment for each element of the globally as well as the locally optimal parental strategies.

## Tangent line of fitness function through the origin can determine optimal parental investment pattern.

By tabulating the properties of *Sex-1*- and *Sex-2*-fitness functions together with the optimal strategies of all models (Table S4), we revealed that the comparison of tangent lines through the origin (henceforth, tangent line for brevity; dotted lines in Figs. 3a, 4a, Figs. S3–S16, S18–S32, panels (a)) for the two types of fitness functions can be utilized to predict the pattern of locally optimal allocation. According to this tangent-line rule of thumb, as *S* increases, the optimally acting parents should first preferentially invest more into the group (sex) of offspring with the steeper tangent line. As *S* further increases, a higher proportion (compared to the condition when *S* is low) of the investment should be given to the sex with the lower tangent line. The switch from >50% investment towards one sex to >50% investment towards the other sex as *S* increases is crucial in validating the predictions from the classical TWH. However, this switch may (e.g., the presence of an intersection for each *1° SR* in Fig. 3d and for *1° SR* = 0.3 and 0.5 in Fig. 4d) or may not (e.g., the absence of an intersection for *1° SR* = 0.7 in Fig. 4d) be observed in locally optimal parental strategies. The pattern of investment allocation (proportion of *S* toward the same-sex offspring) in the globally optimal (optimal *1° SR*) strategies also appears to be generally consistent with the tangent-line rule of thumb (Figs. S19–S32; Table S4). Finally, the tangent-line rule of thumb predicts that the components of locally optimal parental investment strategies in each *1° SR* may follow the classical or reversed TWH pattern. This rule also complies with the observed switch between the classical and reversed TWH trends of investment proportion when the sex with the steeper tangent line switched between models M7-4 and

M7-5 (illustrated in Figs. S10, S11, S17, and in Supplementary Animation S1, S2).

Mathematical analyses of some specific cases also highlight the importance of the sex difference in tangent lines when predicting optimal sex-biased investment (Theorems 4, 5 in SM, Part 4). We speculate that the observed tangent-line rule of thumb is related to the maximum efficiency in terms of fitness divided by the given investment (i.e., $f(x)/x$). We also propose that the tangent-line rule of thumb holds only when *S* is greater than a certain threshold. Suppose that *S* is extremely small, then it is trivial to expect that *Sex 1* should be preferred as $f_1(x) > f_2(x)$ for *x*<*S* due to the definition of two fitness functions. Denote by $a_1^*$ and $a_2^*$ the *x*-values of tangent points (black circles in Figs. 3a, 4a) for fitness functions of *Sex 1* and 2, respectively. We speculate that the larger value of $a_1^*$ and $a_2^*$ (max($a_1^*, a_2^*$)) could be a candidate of this threshold above which tangent rule stays valid (for the rationale of this threshold, see Lemma 6 in SM, Part 4). We hypothesize that the offspring fitness functions in the empirical cases of the reversed TWH pattern[25] might have been similar to those shown in Fig. 4a. Additionally, note that this tangent-line rule of thumb does not universally predict the pattern in the number of the offspring to be cared for (see Remark 3 in SM, Part 4).

## Relevance of the model assumptions and predictions to empirical research.

Two types of data can be collected to verify the validity of the predictions: (1) Empirically derived variables that can be viewed as the *per capita* investment by parents into an offspring of a specific sex ($x_i$), and the fitness accrued by parents from one male and one female as a function of the $x_i$ (i.e., the shape of offspring fitness function); (2) Empirical variables that can be viewed as the index of total investment, primary sex ratio, secondary sex ratio, and allocation of investment between the two sexes in a brood.

One can generally predict optimal patterns using the present model, and compare them with empirical observations. As there is no generally accepted definition of fitness[26,27], empirical papers mentioning fitness often use other alternative and indirect indices such as body size, age at the first reproduction, or longevity[9,28]. Similar issues can arise regarding the measurement of parental investment, offspring-specific or total. For example, the parental investment may concern egg volume/size[29,30], yolk content/volume in an egg[31], hormones in egg's yolk (e.g., androgens)[31], amount of prey brought per offspring to nest[32], or amount of specific type of prey (e.g., large prey for house sparrows[33]). This diversity illustrates the multidimensionality of measuring parental investment[34] as well as the specificity of study organisms and methodological challenges.

If females are highly precocious, while males are highly altricial, then it is expected that the slope of the female tangent line is greater than that of the male tangent line. Suppose that the male fitness function is step-like, and this step (an abrupt escalation of offspring fitness) occurs at relatively low parental investment (low $x_i$). In this case, if the upper bound of the male fitness function is profoundly higher than that of the female fitness function, then the slope of the male tangent line would be accordingly greater. For instance, if inclusion into a group of alpha males ensures high reproductive success, while there is no chance of reproduction otherwise, the offspring fitness function of males would be step-like and profoundly high. Given that such inclusion does not require high amounts of parental investment, the reversed Trivers–Willard hypothesis would be observed in this case. It is notable that counterexamples of the Trivers–Willard hypothesis were found in studies of ungulates whose polygynous social structure is favorable to alpha males[11,35]. The degree of polygyny and corresponding shapes of male and female fitness functions may determine whether classical or reversed TWH is optimal.

In combination with polygyny, the present model could be compatible with the local resource competition (LRC) theory. According to LRC, parents produce more dispersive sex than philopatric sex to reduce competition among offspring of philopatric sex[3]. The variation of local habitat quality would influence the general values of offspring fitness of the philopatric sex, while that of the dispersive sex is less affected. There could be a switch of the tangent lines of the fitness functions from both sexes if the habitat quality changes. Four ungulate species (roe deers, reindeers, Cape mountain zebras, and bighorn sheep) introduced in the review of Hewison and Gaillard[11] exhibited a significant negative relationship between male offspring ratio and higher maternal condition (or ranking). These ungulates are polygynous, or the males thereof are dispersive[36–39]. Fitness functions incorporating social structures of LRC and polygamy may explain patterns of offspring sex ratios.

## Conclusions

While our models cover a variety of sex differences in logistic fitness functions, they simplify some other aspects as discussed in SM, Part 1 including physiological constraints and lack of information. Our focus on the fitness function could incorporate population-level processes[12,16] by modifying the shape of the fitness functions. The simplified theoretical formulation allowed clarity in mathematical, computational, and graphical analyses of the effects of fitness function shape on the comprehensive multi-element parental strategy for broods with multiple offspring. We believe that our models provide a thorough theoretical framework that visualizes how sex differences in fitness functions can lead to a variety of optimal parental strategies, including the classical and the reversed TWH predictions. Although empirical measurements of fitness functions might be difficult[9,27], our results highlight the importance of empirically derived fitness functions of male and female offspring in understanding the optimal parental investment. We propose that the shape difference in fitness functions, rather than the difference in fitness variance/range, should be measured, analyzed, and utilized in future empirical research on parental investment into male and female offspring.

## Methods

### Definitions of terms describing multiple aspects of parental strategy

*Clutch size.* The number of offspring before investment processes started (i.e., pre-investment number of offspring or number of offspring at conception). From the biological perspective, this may be viewed, approximately, as the number of fertilized eggs inside of the female body before nourishment from the mother begins (this ignores the possibility that the eggs may be differentially invested before the fertilization). For avian species, this may also refer to the number of eggs laid by a female bird before parental investment starts (this also ignores the possibility of differential investment into the yolk of each egg).

*Sex.* In this study, we used neutral terms, *Sex 1* and *Sex 2*, to specify the sexes with different fitness functions. This is because it was difficult to determine which fitness function corresponds to that of conventional male or female offspring in many models. In all models, for a pair of two fitness functions, there exists certain $L$ ($L > 1$) of *per capita* investment ($x$) such that $f_1(x) > f_2(x)$ holds for $1 < x < L$ where $f_1, f_2$ are the fitness functions of *Sex 1* and *Sex 2*, respectively. This indicates that the fitness of *Sex 1* is greater than that of *Sex 2* when the given investment is small.

*Primary sex ratio (1° SR).* Sex ratio expressed either as the proportion of *Sex 2* offspring in a brood (e.g., 0.3) or as the ratio of the number of *Sex 1*: the number of *Sex 2* offspring in a brood of 10 (e.g., 7:3) at the stage before the investment begins. From the biological perspective, it is equivalent to the sex ratio at the beginning of the investment.

*Secondary sex ratio (2° SR).* Sex ratio expressed either as the proportion of *Sex 2* offspring in a brood or as the ratio of the number of *Sex 1*: the number of *Sex 2* offspring at the end of parental investment. From the biological perspective, this may be viewed as the brood sex ratio at offspring's independence.

*Care, investment, parental resource allocation.* All terms refer to the amount that parents invest into their offspring.

*Allocation rule.* This involves three elements in the model. One element is the number of offspring chosen for investment, i.e., the number of offspring to which the given investment is larger than zero (see *Brood size* below). The second element is the amount of investment for each offspring chosen for investment. As the optimal solutions turned out to be the equitable investment to offspring within each sex, this boils down to investment for one *Sex 1* or for one *Sex 2* offspring (i.e., *per capita* investment). The final element of the allocation rule is the proportion of total investment ($S$) allocated to all *Sex 1* compared to the proportion of investment to all *Sex 2* in the brood.

*Brood size.* The number of offspring in the model that was chosen for investment larger than zero. The model simplifies the reality as it assumes that the mortality (the abandoned offspring which will have investment ratios equal to zero) associated with parental allocation rules is determined at the initial stage.

**Analytical methods.** Full details of the mathematical analysis including theorems, corollaries, lemmas, and remarks are provided in Supplementary Materials, Part 4. We assumed that the fitness accrued by parents from one offspring, $f_{b_i}$, as a function of parental *per capita* investment ($x_i$) follows logistic function: $f_\kappa(x_i) = \frac{\alpha_\kappa}{1+e^{-\beta_\kappa(x_i-\gamma_\kappa)}} + \delta_\kappa$ where $\kappa = b_i$. The general shape of the sigmoid function is a reasonable approximation for simulating the effect of parental investment on offspring fitness[20]. (However, some may argue that $f_{b_i}(x_i) = 0$ would be more realistic for small range of $x_i$.) In mathematical analysis, we proved that differential values (i.e., values obtained through the differentiation) of fitness functions from offspring that are cared for in the optimized conditions are identical regardless of the fitness curve shape (*Theorem 1*). In other words, $f'_{b_i}(x_i^*)$'s are identical where non-zero $x_i^*$ is the optimized *per capita* investment to the $i$-th offspring. Note that this principle is for offspring which are cared for: that is to say, differential values for offspring which are not cared for in optimized conditions are exempted from this principle. Based on this principle, we proved that the optimal number of offspring to be cared for monotonically increases with the total amount of expendable care if fitness functions are identical.

Where there are two types of fitness functions, it can be also expected that selecting some of the offspring of a given type and providing equitable care is optimal (*Lemma 4*). While we could not find a general analytical method to derive optimal allocation rules with two different types of fitness functions, we instead proposed an algorithm that reduces calculation time compared to the exhaustive search (*Algorithm 1*).

**Computational methods.** Additional discussion of assumptions is presented in Supplementary Materials, Part 1. We simulated broods with 14 different pairs of fitness functions (shown in models *M3* to *M6*, and *M7-1* to *M7-10*), each pair tested in the three different primary brood sex ratios of 0.3, 0.5, and 0.7. The aim was to find the multi-element parental strategy (primary sex ratio, brood size, *per capita* investment, allocation proportion, secondary sex ratio) that maximizes the fitness of the whole brood receiving total parental investment $S$. Let $x_i$ stand for the amount of investment given to the $i$-th offspring (*per capita* investment), then $\sum_{i=1}^{N} x_i = S$ where $N$ is the number of offspring in a clutch ($N = 10$ in the computations). By allowing $x_i = 0$ for some nestlings, we imitated biological processes responsible for brood size reduction. Total offspring fitness $F(X)$, equivalent to total fitness accrued by parents from the brood when expendable investment is $S$, can be defined as $F(X) = \sum_{i=1}^{N} f_{b_i}(x_i)$, where $f_{b_i}$ is the fitness function of $i$-th offspring. Then, we determined the distribution of parental *per capita* investment among offspring (those with $x_i > 0$) and the proportion of resources invested in each sex out of total $S$ (in models simulating two sexes in a brood) that resulted in maximization of $F(X)$. We created various shapes of fitness functions by changing parameters ($\alpha_\kappa, \beta_\kappa, \gamma_\kappa, \delta_\kappa$) in the logistic equation (Table S1).

Parents in decent condition can provide ample prenatal or postnatal care to offspring until their independence. To reduce complexity, we assumed that the maternal condition is proportional to the amount of investment provided to the offspring, and we did not consider the condition of offspring genetically inherited from parents. This concept is embodied by $S$, the total expendable parental investment. Therefore, high $S$ indicates that parents are in a good condition, while the parental condition does not affect the shape of the offspring fitness function.

Additionally, there are constraints on $x_i$'s such that

$$0 \le x_i \le S \text{ for all } i's \text{ and } \sum_{i=1}^{N} x_i = S \tag{1}$$

Let $\mathcal{H}$ denote the set of $N$-tuples satisfying the aforementioned constraints.

$$\mathcal{H} = \{X = (x_1, x_2, x_3, \cdots, x_N) \in \mathbb{R}^N | 0 \le x_i \le S \text{ for } i = 1, 2, 3, \cdots, N \text{ and } \sum_{i=1}^{N} x_i = S\}. \tag{2}$$

The aim of the study is to solve the optimization problem:

$$X^* = (\nu_1, \cdots, \nu_N) \text{ such that } X^* = \underset{X \in \mathcal{H}}{\arg\max} F(X) \text{ where } F(X) = \sum_{i=1}^{N} f_{b_i}(x_i). \tag{3}$$

where $b_i$'s indicate the type of fitness function.

As there are inequality constraints in the domain, it would be appropriate to utilize Karush–Kuhn–Tucker (KKT) conditions to find $X^*$. To solve the problem, however, we compared the fitness of all possible distributions of $X$ for simplicity

rather than using KKT conditions. We used the *fmincon* solver in MATLAB to find multiple local maxima. By comparing combinations of $x_i$'s, it is possible to find $X^*$ that maximizes $F(X^*)$. By looking into the distribution of $X^*$, one can determine how many offspring should be cared for and how investment should be allocated among *Sex 1* and *Sex 2* offspring in order to maximize total parental fitness[40].

In the models where offspring of different sex have different fitness functions (model M3–M7), we analyzed optimal strategies for *Sex-1*-biased, equal, and *Sex-2*-biased conditions. In *Sex-1*-biased brood, for example, 7 *Sex 1* and 3 *Sex 2* offspring are present. In order for parents of *Sex-1*-biased broods to optimize the allocation of the total expendable care, they should maximize

$$\sum_{i=1}^{N} f_{b_i}(x_i) = f_1(x_1) + f_1(x_2) + f_1(x_3) + f_1(x_4)$$
$$+ f_1(x_5) + f_1(x_6) + f_1(x_7) + f_2(x_8) + f_2(x_9) + f_2(x_{10}) \tag{4}$$

with constraints of $\sum_{i=1}^{10} x_i = S$, where $f_1$ is the fitness function of a *Sex 1* offspring and $f_2$ is that of a *Sex 2* offspring. In this scenario, $b_i = 1$ for $1 \le i \le 7$ and $b_i = 2$ for $8 \le i \le 10$.

We parametrically varied the value of *S* from 1 to 1000 (though it may not all be revealed in the graphs) in steps of 1. This allowed us to determine how much care should be given to each offspring in order to maximize total fitness.

**Reporting summary**. Further information on research design is available in the Nature Research Reporting Summary linked to this article.

## Data availability
The datasets which are generated and analyzed in the present study are available at https://doi.org/10.5281/zenodo.5920077.

## Code availability
The details of the fitness functions, the codes, and the results of optimization using MATLAB *fmincon* are available at https://doi.org/10.5281/zenodo.5920077.

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

## Acknowledgements
J.C. was supported by Yangsook Park & Youngho Jung Funds for basics sciences and Basic Science Research Program through the National Research Foundation of Korea (NRF) funded by the Ministry of Education (2021R1A6A3A01086754). M.K. and H.K. were supported by NRF-2021R1A2C1009878 and NRF-2021R1A2C3010887, respectively. This work was also partly supported by the NRF-2019R1A2C1004300, BK21 funded by NRF and the Ministry of Education (Korea, Rep.) to PGJ and DGIST Start-up Fund Program (20200810) of the Ministry of Science, ICT and Future Planning (Korea, Rep.) to S.L.

## Author contributions
Initial design: J.C., S.L., P.G.J.; Programming: J.C.; Mathematical analysis: J.C., H.R., H.K., M.K.; Initial writing: J.C.; Biological interpretation and literature review: S.L., P.G.J.; Revision and review: J.C., S.L., P.G.J.

## Competing interests
The authors declare no competing interests.
