## [Peer Review File · Communications Biology]

Reviewers' comments:

Reviewer #1 (Remarks to the Author):

I find this to be a really nice study that comprehensively demonstrates what our expectations should be for optimal sex allocation under various socio-ecological conditions. It represents a clear improvement on existing theory in this field, which has often only looked at a strongly limited subset of cases (e.g., the focus on males vs. females, where there are as many cases representing exceptions to the rule as adhering to the rule itself). Worse, it is a field that has long been muddled by vague verbal arguments (e.g., offspring with higher "variance in fitness" (variance over what?) should receive more investment "when parents are in high condition" (what is high?). I therefore applaud your efforts in building and analyzing this very general model, whose results aren't all that surprising once the model setup is properly understood – and this is a good thing. My main requests for changes are entirely presentational, and I do think this manuscript has a lot to gain on improving the clarity both of its structure, figures, and language.

Some general comments:

A) The numbering of references is not in the order in which they appear in the manuscript. (After reference 1 we are shown reference 5, then references 11 and 12...) Needs updating!

B) Figure 1 and the Results section is very dense and difficult to navigate, although I appreciate that you've already done a massive job in narrowing all the results down to only the most important ones presented in the paper (c.f. the 50 pages of supplementary material...). I think a major improvement would be if just a bit more work was put in the presentation of the figures, and spending just slightly more time guiding the reader through the results. Specifically, splitting figure 1 with its 16 subpanels into several smaller figures, each with fewer subpanels, might help group certain results together. Many subpanels can be OK if they are all the same type of graph, or if there is a clear thread between them, but here there are very many different graph types, and finding the logical thread connecting them is very challenging. Most of the panels are useful on their own, but when the text introduces the subpanels in a non-chronological way (starting on line 92), and the reader needs to sift through all 16 panels across five rows of a figure to find the one you mention, it becomes unruly. Probably your best bet is to split the figure up into 1) and introductory one-panel figure of panel a; 2) a three-panel figure with panels b-d for model M1; 3) a six-panel figure with panels e-j for model M3; and 4) another six-panel figure with panels k-p for model M5.

Another option is to group similar graph types together, so e.g. one figure with panels b, e & k; one with panels c, f & l, et cetera. I'm sure there are other logical ways of doing this as well. Alternatively, a few more words on the figure in the form of "subheadings" over certain panels/rows might help. E.g., you state in the beginning of the figure legend that the figure shows "three examples chosen from the models presented in Supp Mat...", but it is not at all clear from the figure what panels belong to which examples, or what the examples even are. It's difficult even after digging through the lengthy figure legend, and the main text. Some simple visual formatting could help it be clear from first glance, for example the subheadings "Model M1: Fitness functions identical for both sexes" for panels b-d, and something like "Model M3: Low-variance sex requires little investment" and "Model M5: Low-variance sex requires more investment" (or something more accurate) respectively for panels e-j and k-p.

C) I think just a little bit more discussion would strengthen the paper considerably. I understand that journal word limits might be at play, but the paper is currently quite short and I think it would really help empiricists interpret the importance of your work if you zoomed out a bit more at the end after finishing the technical bits. Currently the "Tangent line rule" is the last subheading in your "Results and discussion" section, but the final paragraph is clearly more general and could have a "Conclusions" subheading or something. It's good that you briefly address shortcomings/limitations of your model, but I also think you should mention some parts that are more relevant to empiricists rather than just modelers, as the journal is quite wide in scope. For example, what biological situations (ecological or social conditions) would perhaps create the different kinds of fitness functions you model (M3-M7)? When would the logistic curves be steeper or shallower, have earlier or later inflection points, have different upper asymptotes? When and why would sex differences in these arise? See an example of similar discussion in Haaland et al.

2017 Am Nat: <https://doi.org/10.1086/693484>. Can you find examples in the published studies reporting a classical or reversed TWH pattern where one of your model scenarios might explain the results? It doesn't have to be a comprehensive search; just a few cases illustrating how your model could be applied may go a long way.

Below is a rather long list of mostly nitpicky edits of style, grammar and wording. I hope you forgive me for this and realize that I did it because I really like the paper, consider it highly publishable, and therefore want it to be as clear and readable as possible in all aspects.

Specific comments:

Line 28: Add "the": "The Trivers-Willard hypothesis (TWH)..."

Line 31: Secondary sex ratios are an outcome of the parental strategy, not actually part of it, right? Perhaps add "resulting". Also, "The results" -> "Our results". "... and resulting secondary sex ratios. Our results reveal..."

Line 35: Comms Biol guidelines specifically states to avoid words such as "novel". Perhaps reword slightly, e.g. "This comprehensive reformulation of TWH offers a thorough understanding..."

Line 46: "(range of fitness function)": You haven't defined this function yet. Also, the historical use of "which offspring type provides the largest fitness variance" is quite vague, maybe you can highlight that here, since a strength of your model is that it offers a much more accurate criterion. So, maybe change this to something like "(typically defined as the range of offspring fitnesses as a function of parental investment)"?

Line 51: Add "the": "the Trivers-Willard hypothesis (TWH)..."

Line 53: Some missing plurals: "sex-specific costs of rearing, sex-specific growth curves,..."

Line 55: Add "has": "While the theory has brought about..."

Line 56: Change "or the" -> "and" to avoid that the two elements in the parenthesis sound like the same. "...primary sex ratio and sex-biased provisioning...". Also, "they" -> "these have", since you're referring to the empirical studies. "...these have produced..."

Line 57: Contradicting -> contrasting. The empirically observed parental strategies differ, they don't contradict each other. (Models or predictions may contradict each other.) Also, missing "have". "...the contrasting parental strategies, we have built..."

Line 60: Remove some superfluous words: "...for the understanding of the variation..." -> "...for understanding the variation..."

Line 64: Slightly strange wording, small suggested removal and rewrite: "However, the typical models presented so far did not..." -> "However, existing models do not..."

Line 67: Small grammar and wording changes. A missing s, an extra s, and specify that you're talking about offspring fitness functions: "...sex differences in the logistic (sigmoid) offspring fitness functions affect the..."

Line 89: This subheading doesn't relate to the paragraph below it, which is a general introduction to the results section. Rather, move the subheading down one paragraph, to between line 94 and 95.

Line 98: Remove a "the": "...the number of the offspring..." -> "...the number of offspring..."

Line 100: Maybe change "are relevant to" to "comply with" or "adhere to"?

Line 108: "...three types of the clutch..." -> "...three types of clutches..."

Line 121: Slightly imprecise to say "maximum fitness is reached with relatively small investment...". I understand what you're trying to describe, but it's not quite correct because in M5, maximum fitness is reached later in Sex 1 than in Sex 2. What you write in the next sentence is more accurate (Sex 2 has a "slower initial increase"), so I suggest to change the wording to contrast more directly with that: "...assumes that fitness increases rapidly with relatively small investment..."

Line 137: The middle part of this sentence, "the optimal primary SR was Sex 2-biased for the larger values of S (>89)", should also have a figure reference, since you're no longer talking about Fig. 11. Which panel is this shown in - 1m? Add reference, and also specify exactly what element of the panel you're talking about (as you nicely do for the previous result, "from 0 to 55; overlapping lines in Fig. 11").

Line 155: Remove a "the", add an "s", change wording: "...indicating the interactive selection pressure..." -> "...indicating interacting selection pressures...". (Interactive means something one can interact with.)

Line 163: Change "Sex 1 group over Sex 2 group" -> "Sex 1 offspring over Sex 2 offspring". You haven't really introduced calling Sex 1 and Sex 2 "groups" before... seems conceptually simpler to

just stick to "offspring" here.

Line 164: Maybe add specification of where we're looking: "orange-shaded plot in Fig. 1h". Also on line 165: "blue line in Fig. 1g".

Line 175: Maybe change "the optimal parents" -> "optimally acting parents"

Line 177 & 179: Maybe change "inversion" -> "switch"

Line 178: "an" -> "the". "...investment towards the other sex..."

Line 179 & 180: Missing article two places: "...e.g. the presence/absence of an intersection..."

Line 182: Talking about both group and sex seems unnecessary? Removing "group" also avoids unfortunate double parenthesis... "(proportion of S toward each group (sex))" -> "(proportion of S toward each sex)"

Line 186: Slightly imprecise wording: Change "...when the sex of the greater tangent line..." -> "...when the sex with the steeper tangent line..."

Line 191: Slightly imprecise wording: Change "...maximum efficiency of fitness against a given investment..." -> "...maximum efficiency in terms of fitness gained per investment..."

Line 198: "...similar to Fig. 1k" -> "...similar to those shown in Fig. 1k".

Line 206: Remove s: "fitness function shape"

Line 217: "The definitions" -> "Definitions".

Line 224: "to specify the sex" -> "to specify the sexes"

Line 247: Some grammar changes, remove several words: "The model simplifies the reality as it is assumed in the model that the mortality..." -> "The model simplifies reality as it assumes that mortality..."

Line 257: I'm not sure what "differential values of fitness functions" means. Do you mean "different fitness function parameter values"? Some rewriting needed.

Line 261: "...number of cared offspring..." -> "number of offspring to be cared for..."

Line 264: "...some of the offspring within the type..." -> "...some of the offspring of a given type..."

Line 265: By "optimal distributions" (plural s needed), do you mean "optimal allocation rules"? If so, change.

Line 271: Perhaps specify a bit better what you're talking about here by adding model names and actually stating what the sex ratios are rather than just saying "the three" sex ratio. "We simulated broods with 14 different pairs of fitness functions (termed models M3 through M6 and M7-1 to M7-10), each pair tested in three different primary brood sex ratios: 0.3, 0.5 and 0.7."

Line 276: Remove "the" before "brood size reduction".

Line 277: Remove starting parenthesis before "F(X)", as this one never gets a closing parenthesis, and it's not needed.

Line 278: Misplaced comma. Remove comma before "F(X)", add comma after the summation term. "...defined as $F(X) = \sum f_b(x_i)$, where..."

Line 285: Remove "a" before "decent condition". Also, is this an assumption you bake in to the calculation? Or just a statement justifying the lack of an offspring-quality effect at independence? In general it is unclear what point this paragraph is making.

Line 286: Change "...is in proportion to..." -> "...is proportional to..."

Line 291: Change "provision" -> "investment". You hardly talk about provisioning at all, investment is much more general.

Line 297: Change "Let denote H..." -> "Let H denote..."

Line 301-305: Formatting and punctuation makes it unclear what is happening here. Perhaps change the period on line 301 to a colon, and remove "(OP) To find". The abbreviation "OP" (for optimization problem?) is never returned to again and thus unnecessary. So you get the structure "The aim of the study is to solve the optimization problem:

$X^* = (v_1, \dots, v_N)$ such that $X^* = \operatorname{argmax} F(X)$ where $F(X) = \sum f_{b_i}(x_i)$

where b_i indicate-s the type of fitness function."

(Note an added "s" and a removed "the" on line 305.

Line 308: Change "We used fmincon of MATLAB..." -> "We used the fmincon function in MATLAB..."

Line 309: Change "By comparing possible combinations x_i 's" -> "By comparing different combinations of x_i 's..."

Line 310-312: Rewrite slightly for precision and readability. "...how many offspring should be cared for, and how investment should be allocated among Sex 1 and Sex 2 offspring in order to maximize total parental fitness."

Line 315-317: Rewrite slightly for precision, grammar and readability. "In the models where offspring of different sex have different fitness functions (M3-M7), we analyzed optimal strategies

for Sex-1-biased, equal, and Sex-2-biased broods. In Sex-1-biased broods, ...".

Line: 318: Small suggested changes for grammar and precision: "In order for parents of Sex-1-biased broods to optimize allocation of the total expendable care...".

Line 322: Add missing comma and change "a" -> "the". "...sum(x_i) = S, where f₁ is the fitness function...".

Line 323: Change "condition" -> "scenario".

Line 326: Missing s. "Sex-2-biased broods"-

Line 327: Remove "of a": "each pair of fitness functions". Change "constrained resources" -> "resource constraints".

Figure legend:

Line 342: Missing s: "...identical fitness functions".

Line 344: "...the number of the cared offspring" -> "...the number of offspring receiving care".

Line 345: Not capital R in "relationship". Remove "the" before "per-capita".

Line 346: As in your legend for panel b, spend just a few words telling the reader what the scenario is. "...Model M3, in which xxx...".

Line 347: Perhaps remove "Based on the upper bound of the functions, Sex 1 corresponds to the female and Sex 2 to the male." This seems unnecessarily limiting, and not something you rely on elsewhere.

Line 349: When describing this panel f, you also need to specify what the colored bars at the top of the panel represent. (The text in the figure itself is a bit ambiguous.)

Line 352: Capital T in "The relationship".

Line 359: "...for model M5": Again, explain briefly what differs here from model M3 above (in terms of scenario setup, not just the results. "...in which the reversed TWH predictions are observed": You can also spend a few more words describing where exactly these results are seen. For example something like: "E.g., Sex 1 offspring (red lines) receive most investment at high parental energetic states for equal or Sex-2-biased primary sex ratios, despite having a lower variance in fitness (n-p).".

Supplementary material: I haven't read the Supplementary Material in extensive detail, only used it for reference when needed while working through the main text, but in general it seems highly satisfactory. One suggestion for improvement may be to add a list of the numbered figures and tables (and possibly also theorems, lemmas and proofs?) with page numbers in or just after the table of contents? Such a list could also include a brief description ("header" for each element). This would make looking for a certain figure or table much easier.

Reviewer #2 (Remarks to the Author):

This study builds on the classic Trivers-Willard model of sex ratio and investment in animals by developing additional models to test this classic idea. The authors focus specifically on the role that different fitness functions (S) have on offspring investment and sex ratio. To me there is a bit of a disconnect between this approach and that of the TWH. The TWH argues that reproductive variance is key to parental sex investment. This study argues that the slope of the fitness function is key. They broach the idea of reproductive variance briefly, but I had trouble following how this is incorporated into the models. To basically argue that the fitness function of offspring is more important than the reproductive variance for each sex (as is done in the abstract, but nowhere else in the paper, the model has to explicitly address reproductive variance.

More generally, it would help if the paper was written with a clearer set of predictions (and how those differ from the TWH). The Discussion sort of ends abruptly in the current version and could be expanded to more clearly compare and contrast the results of this study with TWH in more detail, as well other models on parental investment theory, clutch ratio evolution, etc. As it stands this manuscript covers a lot of ground but could do better integrating with the published literature and more clearly stating the important results in that context.

Specific comments

L33. Unclear what this means in a practical sense.

L45. The key prediction of the TWH is about investment as a function of reproductive variance.

L73. What happens if you use a diminishing returns curve for S instead of sigmoidal function?

L89. It is unclear whether in this section, nests are composed of only one sex.

L95. Do model results change if the clutch size changes (10 is quite high). How much of this result is simply due to producing so many offspring?

L100. This section has nothing to do with the TWH. It has to do with clutch size evolution, and seems out of place given the Introduction.

L128. Your abstract argues that these functions are more important than reproductive variance, but this section states otherwise.

L131. Isn't this argument just one of reproductive value? In other words, parents invest in chicks with higher reproductive value (S).

L162. But what does this mean in practical terms?

Response to Reviewers' Comments

We sincerely appreciate the endeavor and time of two anonymous reviewers that have been spent on comments and scrutiny of the manuscript. Due to considerate analysis and evaluation performed by the reviewers, we were able to find glitches or missing points, enabling us to augment the logical structure of the manuscript. The comments also improved the clarity of the paragraphs and visual data. We incorporated the comments from the reviewers into the main article and supplementary materials. As an important improvement, we added the possible empirical method that can corroborate the theoretical findings. The followings are the point-by-point response to the comments from the reviewers. We numbered the comments for smooth communication about our responses in the further process.

Comment: 1: The numbering of references is not in the order in which they appear in the manuscript. (After reference 1 we are shown reference 5, then references 11 and 12...) Needs updating!

Response: We updated the reference as pointed out by the reviewer.

Comment 2: Figure 1 and the Results section is very dense and difficult to navigate, although I appreciate that you've already done a massive job in narrowing all the results down to only the most important ones presented in the paper (c.f. the 50 pages of supplementary material...). I think a major improvement would be if just a bit more work was put in the presentation of the figures, and spending just slightly more time guiding the reader through the results. Specifically, splitting figure 1 with its 16 subpanels into several smaller figures, each with fewer subpanels, might help group certain results together. Many subpanels can be OK if they are all the same type of graph, or if there is a clear thread between them, but here there are very many different graph types, and finding the logical thread connecting them is very challenging. Most of the panels are useful on their own, but when the text introduces the subpanels in a non-chronological way (starting on line 92), and the reader needs to sift through all 16 panels across five rows of a figure to find the one you mention, it becomes unruly. Probably your best bet is to split the figure up into 1) and introductory one-panel figure of panel a; 2) a three-panel figure with panels b-d for model M1; 3) a six-panel figure with panels e-j for model M3; and 4) another six-panel figure with panels k-p for model M5.

Another option is to group similar graph types together, so e.g. one figure with panels b, e & k; one with panels c, f & l, et cetera. I'm sure there are other logical ways of doing this as well. Alternatively, a few more words on the figure in the form of "subheadings" over certain panels/rows might help. E.g., you state in the beginning of the figure legend that the figure shows "three examples chosen from the models presented in Supp Mat...", but it is not at all clear from the figure what panels belong to which examples, or what the examples even are. It's difficult even after digging through the lengthy figure legend, and the main text. Some simple visual formatting could help it be clear from first glance, for example the subheadings "Model M1: Fitness functions identical for both sexes" for panels b-d, and something like "Model M3: Low-variance sex requires little investment" and "Model M5: Low-variance sex requires more investment" (or something more accurate) respectively for panels e-j and k-p.

Response: In accordance with the suggestions by the reviewer, we divided the previous Figure 1 into four separate figures. We also added additional explanations to make them more clear with modified captions.

Comment 3: I think just a little bit more discussion would strengthen the paper considerably. I understand that journal word limits might be at play, but the paper is currently quite short and I think it would really help empiricists interpret the importance of your work if you zoomed out a bit more at the end after finishing the technical bits. Currently the “Tangent line rule” is the last subheading in your “Results and discussion” section, but the final paragraph is clearly more general and could have a “Conclusions” subheading or something. It’s good that you briefly address shortcomings/limitations of your model, but I also think you should mention some parts that are more relevant to empiricists rather than just modelers, as the journal is quite wide in scope. For example, what biological situations (ecological or social conditions) would perhaps create the different kinds of fitness functions you model (M3-M7)? When would the logistic curves be steeper or shallower, have earlier or later inflection points, have different upper asymptotes? When and why would sex differences in these arise? See an example of similar discussion in Haaland et al. 2017 *Am Nat*: <https://doi.org/10.1086/693484>. Can you find examples in the published studies reporting a classical or reversed TWH pattern where one of your model scenarios might explain the results? It doesn’t have to be a comprehensive search; just a few cases illustrating how your model could be applied may go a long way.

Response: Inspired by the reviewer’s comments, we have added a discussion of links between the model and published empirical research in a new part “Relevance of the model assumptions and predictions to empirical research.” We also added the subheading “Conclusions” before the concluding paragraph, as suggested by the Reviewer.

Comment 4: Line 28: Add “the”: “The Trivers-Willard hypothesis (TWH)...”

Response: We added the “the” as pointed by the reviewer.

Comment 5: Line 31: Secondary sex ratios are an outcome of the parental strategy, not actually part of it, right? Perhaps add “resulting”. Also, “The results” -> “Our results”. “... and resulting secondary sex ratios. Our results reveal...”.

Response: We added the “resultant” in front of the secondary sex ratio, and we changed “The results” to “Our results.”

Comment 6: Line 35: *Comms Biol* guidelines specifically states to avoid words such as “novel”. Perhaps reword slightly, e.g. “This comprehensive reformulation of TWH offers a thorough understanding...”.

Response: We changed “novel” to “thorough.”

Comment 7: Line 46: “(range of fitness function)”: You haven’t defined this function yet. Also, the

historical use of “which offspring type provides the largest fitness variance” is quite vague, maybe you can highlight that here, since a strength of your model is that it offers a much more accurate criterion. So, maybe change this to something like “(typically defined as the range of offspring fitnesses as a function of parental investment)”?

Response: In accordance with the comment, we have rewritten this part to avoid this issue. We changed the sentence and added the definition of the variance of the fitness function as pointed out by the reviewer. We also elucidated the concept of the fitness function at its first appearance.

Comment 8: Line 51: Add “the”: “the Trivers-Willard hypothesis (TWH)...”.

Response: We added the “the” as pointed by the reviewer.

Comment 9: Line 53: Some missing plurals: “sex-specific costs of rearing, sex-specific growth curves,..”.

Response: We changed the nouns into plural forms.

Comment 10: Line 55: Add “has”: “While the theory has brought about...”.

Response: We added “has” as pointed out by the reviewer.

Comment 11: Line 56: Change “or the” -> “and” to avoid that the two elements in the parenthesis sound like the same. “...primary sex ratio and sex-biased provisioning...”. Also, “they” -> “these have”, since you’re referring to the empirical studies. “...these have produced...”.

Response: We removed “or the” and “they” and changed them as pointed out by the reviewer.

Comment 12: Line 57: Contradicting -> contrasting. The empirically observed parental strategies differ, they don’t contradict each other. (Models or predictions may contradict each other.) Also, missing “have”. “...the contrasting parental strategies, we have built...”

Response: We revised the sentence as pointed out by the reviewer.

Comment 13: Line 60: Remove some superfluous words: “...for the understanding of the variation...” -> “...for understanding the variation...”.

Response: We removed the redundant words to promote readability as pointed out by the reviewer.

Comment 14: Line 64: Slightly strange wording, small suggested removal and rewrite: “However, the typical models presented so far did not...” -> “However, existing models do not...”

Response: We simplified the sentence as pointed out by the reviewer.

Comment 15: Line 67: Small grammar and wording changes. A missing s, an extra s, and specify that you’re talking about offspring fitness functions: “...sex differences in the logistic (sigmoid) offspring fitness functions affect the...”

Response: We changed the sentence as pointed out by the reviewer.

Comment 16: Line 89: This subheading doesn't relate to the paragraph below it, which is a general introduction to the results section. Rather, move the subheading down one paragraph, to between line 94 and 95.

Response: We moved the position of the subheading as pointed out by the reviewer.

Comment 17: Line 98: Remove a "the": "...the number of the offspring..." -> "...the number of offspring...".

Response: We removed the "the" as pointed out by the reviewer.

Comment 18: Line 100: Maybe change "are relevant to" to "comply with" or "adhere to"?

Response: We changed "are relevant to" to "comply with" as pointed out by the reviewer.

Comment 19: Line 108: "...three types of the clutch..." -> "...three types of clutches..."

Response: We changed the sentence as pointed out by the reviewer.

Comment 20: Line 121: Slightly imprecise to say "maximum fitness is reached with relatively small investment...". I understand what you're trying to describe, but it's not quite correct because in M5, maximum fitness is reached later in Sex 1 than in Sex 2. What you write in the next sentence is more accurate (Sex 2 has a "slower initial increase"), so I suggest to change the wording to contrast more directly with that: "...assumes that fitness increases rapidly with relatively small investment...".

Response: We modified the text as suggested by the reviewer. Please have a look at pg. 4, lines 12-17.

Comment 21: Line 137: The middle part of this sentence, "the optimal primary SR was Sex 2-biased for the larger values of S (>89)", should also have a figure reference, since you're no longer talking about Fig. 1l. Which panel is this shown in – 1m? Add reference, and also specify exactly what element of the panel you're talking about (as you nicely do for the previous result, "from 0 to 55; overlapping lines in Fig. 1l").

Response: Optimal primary sex ratio was determined by comparing the "fitness from optimal brood" in the original Figure 1l (which is the revised Fig. 4b). Three horizontal bars on the top of the Fig. 1l and 1m (revised Fig. 4b, c) indicate ranges of investment (S) for which particular primary sex ratio(s) have the highest value of the "fitness from optimal brood." This is done by determining which of the three functions in Fig.1l (revised Fig. 4b) have the highest value. In accordance with the suggestion here, and with the suggestions in Comment 2, we added the explanation and figure reference to this sentence.

Comment 22: Line 155: Remove a "the", add an "s", change wording: "...indicating the interactive selection pressure..." -> "...indicating interacting selection pressures...". (Interactive means something one can interact with.)

Response: We modified the sentence in accordance with the reviewer's suggestion.

Comment 23: Line 163: Change “Sex 1 group over Sex 2 group” -> “Sex 1 offspring over Sex 2 offspring”. You haven't really introduced calling Sex 1 and Sex 2 “groups” before... seems conceptually simpler to just stick to “offspring” here.

Response: In accordance with the Reviewer's comment, we have modified the text. The revised version is: “We used the term “group” to indicate that we compared the summation of investments that goes to all offspring of the same sex. To reduce ambiguity, “Sex 1 group” and “Sex 2 group” are changed to “whole Sex 1 offspring” and “whole Sex 2 offspring,” respectively. Please have a look at pg. 5, lines 15-18.

Comment 24: Line 164: Maybe add specification of where we're looking: “orange-shaded plot in Fig. 1h”. Also on line 165: “blue line in Fig. 1g”.

Response: Considering this comment and in accordance with Comment 2, we have expanded the text that leads the reader through elements of the figures. We added a specific explanation about the points or ranges in the figure that should be noted. Please have a look at line lines pg. 5, 15-25.

Comment 25: Line 175: Maybe change “the optimal parents” -> “optimally acting parents”

Response: We changed the phrase as suggested by the reviewer.

Comment 26: Line 177 & 179: Maybe change “inversion” -> “switch”

Response: We changed the “inversion” to “switch” as suggested by the reviewer.

Comment 27: Line 178: “an” -> “the”. “...investment towards the other sex...”.

Response: We changed the sentence as pointed out by the reviewer.

Comment 28: Line 179 & 180: Missing article two places: “...e.g. the presence/absence of an intersection...”

Response: We added the article as pointed out by the reviewer.

Comment 29: Line 182: Talking about both group and sex seems unnecessary? Removing “group” also avoids unfortunate double parenthesis... “(proportion of S toward each group (sex))” -> “(proportion of S toward each sex)”

Response: In accordance with the Reviewers' comment, we changed “each group (sex)” to “the same-sex offspring” to reduce the redundancy. The following is the revised text: “... The pattern of investment allocation (proportion of S toward the same-sex offspring) in the globally optimal ...” Please have a look at lines pg. 6, lines 2-4.

Comment 30: Line 186: Slightly imprecise wording: Change “...when the sex of the greater tangent line...” -> “...when the sex with the steeper tangent line...”.

Response: We modified the sentence as suggested by the reviewer.

Comment 31: Line 191: Slightly imprecise wording: Change "...maximum efficiency of fitness against a given investment..." -> "...maximum efficiency in terms of fitness gained per investment...".

Response: We modified the phrase to "maximum efficiency in terms of fitness divided by the given investment" to increase clarity. Please see pg. 6, lines 10-12.

Comment 32: Line 198: "...similar to Fig. 1k" -> "...similar to those shown in Fig. 1k".

Response: We changed the sentence as suggested by the reviewer.

Comment 33: Line 206: Remove s: "fitness function shape"

Response: We removed "s" as pointed out by the reviewer.

Comment 34: Line 217: "The definitions" -> "Definitions".

Response: We removed the "The" as pointed out by the reviewer.

Comment 35: Line 224: "to specify the sex" -> "to specify the sexes"

Response: We changed the form of the noun as pointed out by the reviewer.

Comment 36: Line 247: Some grammar changes, remove several words: "The model simplifies the reality as it is assumed in the model that the mortality..." -> "The model simplifies reality as it assumes that mortality..."

Response: We simplified the sentence as pointed out by the reviewer. Please see pg. 8, lines 20-22.

Comment 37: Line 257: I'm not sure what "differential values of fitness functions" means. Do you mean "different fitness function parameter values"? Some rewriting needed.

Response: We meant the differentiation of the function (df/dx), and we modified the text to make it clear. Please see pg. 8, lines 30-32.

Comment 38: Line 261: "...number of cared offspring..." -> "number of offspring to be cared for...".

Response: In accordance with the suggestion, we have changed the expression in the revised manuscript.

Comment 39: Line 264: "...some of the offspring within the type..." -> "...some of the offspring of a given type..."

Response: We changed the sentence as pointed out by the reviewer.

Comment 40: Line 265: By "optimal distributions" (plural s needed), do you mean "optimal allocation rules"? If so, change.

Response: We have modified the sentence in accordance with the suggestion of the reviewer. Please

see pg. 8, lines 38-40.

Comment 41: Line 271: Perhaps specify a bit better what you're talking about here by adding model names and actually stating what the sex ratios are rather than just saying "the three" sex ratio. "We simulated broods with 14 different pairs of fitness functions (termed models M3 through M6 and M7-1 to M7-10), each pair tested in three different primary brood sex ratios: 0.3, 0.5 and 0.7."

Response: We specified the information as pointed out by the reviewer. Please see pg. 8, lines 44-46.

Comment 42: Line 276: Remove "the" before "brood size reduction".

Response: We removed the "the" as pointed out by the reviewer.

Comment 43: Line 277: Remove starting parenthesis before "F(X)", as this one never gets a closing parenthesis, and it's not needed.

Response: We fixed the sentence as pointed out by the reviewer.

Comment 44: Line 278: Misplaced comma. Remove comma before "F(X)", add comma after the summation term. "...defined as $F(X) = \sum f_b(x_i)$, where..."

Response: We moved the comma as pointed out by the reviewer.

Comment 45: Line 285: Remove "a" before "decent condition". Also, is this an assumption you bake in to the calculation? Or just a statement justifying the lack of an offspring-quality effect at independence? In general it is unclear what point this paragraph is making.

Response: The intention of the sentence is that parents in a good condition can provide higher parental investment (S) to the offspring, while we did not consider the effect of genetic inheritance from the parents. We added a sentence at the end of the section to make this clearer: "Therefore, high S indicates that parents are in a good condition, while the parental condition does not affect the shape of the offspring fitness function." Please have a look at pg. 9, lines 5-6.

Comment 46: Line 286: Change "...is in proportion to..." -> "...is proportional to..."

Response: We changed the sentence as pointed out by the reviewer.

Comment 47: Line 291: Change "provision" -> "investment". You hardly talk about provisioning at all, investment is much more general.

Response: We changed the term as pointed out by the reviewer.

Comment 48: Line 297: Change "Let denote H..." -> "Let H denote..."

Response: We fixed the sentence as pointed out by the reviewer.

Comment 49: Line 301-305: Formatting and punctuation makes it unclear what is happening here. Perhaps change the period on line 301 to a colon, and remove "(OP) To find". The abbreviation "OP"

(for optimization problem?) is never returned to again and thus unnecessary. So you get the structure
“The aim of the study is to solve the optimization problem:

$X^* = (v_1, \dots, v_N)$ such that $X^* = \operatorname{argmax} F(X)$ where $F(X) = \sum f_{b_i}(x_i)$

where b_i indicates the type of fitness function.”

(Note an added “s” and a removed “the” on line 305.)

Response: We changed the sentences as pointed out by the reviewer.

Comment 50: Line 308: Change “We used fmincon of MATLAB...” -> “We used the fmincon function in MATLAB...”.

Response: We changed the sentence as pointed out by the reviewer.

Comment 51: Line 309: Change “By comparing possible combinations x_i ’s” -> “By comparing different combinations of x_i ’s...”.

Response: We changed the sentence as pointed out by the reviewer.

Comment 52: Line 310-312: Rewrite slightly for precision and readability. “...how many offspring should be cared for, and how investment should be allocated among Sex 1 and Sex 2 offspring in order to maximize total parental fitness.”.

Response: We changed the sentence in accordance with the comment from the reviewer. Please see pg. 9, lines 28-29.

Comment 53: Line 315-317: Rewrite slightly for precision, grammar and readability. “In the models where offspring of different sex have different fitness functions (M3-M7), we analyzed optimal strategies for Sex-1-biased, equal, and Sex-2-biased broods. In Sex-1-biased broods, ...”.

Response: We modified the sentence as suggested by the reviewer. Please see pg. 8, lines 33-35.

Comment 54: Line: 318: Small suggested changes for grammar and precision: “In order for parents of Sex-1-biased broods to optimize allocation of the total expendable care...”.

Response: We modified the sentence as suggested by the reviewer.

Comment 55: Line 322: Add missing comma and change “a” -> “the”. “... $\sum(x_i) = S$, where f_1 is the fitness function...”.

Response: We changed the sentence as pointed out by the reviewer.

Comment 56: Line 323: Change “condition” -> “scenario”.

Response: We changed the term as pointed out by the reviewer.

Comment 57: Line 326: Missing s. “Sex-2-biased broods”-

Response: We added the “s” as pointed out by the reviewer.

Comment 58: Line 327: Remove “of a”: “each pair of fitness functions”. Change “constrained resources” -> “resource constraints”.

Response: We changed the sentence as pointed out by the reviewer.

Figure legend:

Comment 59: Line 342: Missing s: “...identical fitness functions”.

Response: We added the “s” as pointed out by the reviewer.

Comment 60: Line 344: “...the number of the cared offspring” -> “...the number of offspring receiving care”.

Response: We changed the sentence as pointed out by the reviewer.

Comment 61: Line 345: Not capital R in “relationship”. Remove “the” before “per-capita”.

Response: We fixed the error as pointed out by the reviewer.

Comment 62: Line 346: As in your legend for panel b, spend just a few words telling the reader what the scenario is. “...Model M3, in which xxx...”.

Response: In accordance with Comment 2, we split the figure into 4 separate figures. In the new captions of each figure, we expanded the explanations in accordance with the reviewers' comments.

Comment 63: Line 347: Perhaps remove “Based on the upper bound of the functions, Sex 1 corresponds to the female and Sex 2 to the male.” This seems unnecessarily limiting, and not something you rely on elsewhere.

Response: We removed the sentence as suggested by the reviewer.

Comment 64: Line 349: When describing this panel f, you also need to specify what the colored bars at the top of the panel represent. (The text in the figure itself is a bit ambiguous.)

Response: We added the explanation about the horizontal bars in the caption, and we modified the figure to make it clear.

Comment 65: Line 352: Capital T in “The relationship”.

Response: We fixed the error as pointed out by the reviewer.

Comment 66: Line 359: “...for model M5”: Again, explain briefly what differs here from model M3 above (in terms of scenario setup, not just the results. “...in which the reversed TWH predictions are observed”: You can also spend a few more words describing where exactly these results are seen. For example something like: “E.g., Sex 1 offspring (red lines) receive most investment at high parental energetic states for equal or Sex-2-biased primary sex ratios, despite having a lower variance in fitness (n-p).”.

Response: In accordance with Comment 2, we have split the figure into 4 separate figures and in the new captions of the figures we have expanded the explanations in accordance with the reviewers' comments.

Reviewer 2

Comment 67: L33. Unclear what this means in a practical sense.

Response: We have added a sentence to the abstract to reflect the changes in the revised manuscript that includes an expanded discussion of links between the theoretical model and the empirical research that addresses the issue of the meaning "in a practical sense." Due to the brevity of the abstract, however, we could not add much more to the abstract. We added in the latter part of the Results and Discussion section what these theoretical and model results indicate in the empirical sense.

Comment 68: L45. The key prediction of the TWH is about investment as a function of reproductive variance.

Response: In accordance with the comment, we modified the explanations in the Introduction to reduce the ambiguity.

Comment 69: L73. What happens if you use a diminishing returns curve for S instead of sigmoidal function?

Response: If the fitness is not monotonically increasing but decreases though S increases, then it is optimal not to spend all expendable care on the offspring. If the diminishing return is defined as decreasing but not the negative return, then the sigmoidal function satisfies this condition. This is because there is an upper bound of the function, and the slope becomes flat as the argument of the function increases.

Comment 70: L89. It is unclear whether in this section, nests are composed of only one sex.

Response: We admit that there was inconsistency between the section name and its contents. We moved the section header. Please see pg. 3, line 21.

Comment 71: L95. Do model results change if the clutch size changes (10 is quite high). How much of this result is simply due to producing so many offspring?

Response: The number of the offspring, only if it is not singular, does not affect the general results of the model. Theorem 1 and 2 of Supplementary Materials show that selecting some of the offspring and providing equitable investment to them is optimal, regardless of the number of the offspring. In addition, the number of offspring receiving care should increase with expendable investment (S). Therefore, the pattern shown in Fig. 1c will hold, and the change of the clutch size only will change the upper bound of the number of offspring receiving investment. We have mentioned this in the revised manuscript. Please have a look at pg. 3, lines 26-28.

Comment 72: L100. This section has nothing to do with the TWH. It has to do with clutch size evolution, and seems out of place given the Introduction.

Response: This section provides the basic structure of how the model works, i.e. the situation when all fitness functions are identical. The next step is the expansion of this model to two-sex broods. Also, this model can be viewed as a special situation when males and females have identical fitness functions. As mentioned by the reviewer, the discussion about the brood reduction hypothesis may at first appear additional and not in the same line with the Trivers-Willard hypothesis. However, we believe that this argument is in line with the general logic behind the Trivers-Willard hypothesis in that both theories of brood reduction and sex-biased allocation concern optimal parental allocation strategy into individual offspring.

Comment 73: L128. Your abstract argues that these functions are more important than reproductive variance, but this section states otherwise.

Response: The sentence pointed out by the reviewer is about the model *M3*, in which the predictions follow the traditional TWH reasoning, also consistent with the “tangent-line rule of thumb.” Additionally, we claim that the tangent line of the fitness function determines the optimal allocation rules. If the tangent of the Sex 1 is greater than that of Sex 2, then expectations from the classical Trivers-Willard hypothesis holds. If the tangent of the Sex 1 is smaller than Sex 2, then expectations from the classical Trivers-Willard hypothesis do not hold, and this is the reason we claim that the shape of the fitness function is the crucial factor. Table S4 of Supplementary Materials shows that it should be the tangent, rather than reproductive variance, that should be used to predict optimal parental strategy.

Comment 74: L131. Isn't this argument just one of reproductive value? In other words, parents invest in chicks with higher reproductive value (*S*).

Response: *S*, on the horizontal axis, is not reproductive value. *S* is the total investment that is being allocated among male and female offspring in a brood. The reproductive value is represented on the vertical axis in panels (a) in Fig. 3 and 4 presenting “fitness functions”. To avoid this type of confusion we have rewritten this part to: “Sex-1-biased care for the parents with small *S* and Sex-2-biased care for those with large *S* as evidenced in Fig. 3d.” Please see pg. 4, lines 22-24.

Comment 75: L162. But what does this mean in practical terms?

Response: Reviewer 1 also pointed out that more discussion of links between model and empirical studies is required. We provided a new part of the discussion entitled “Relevance of the model assumptions and predictions to empirical research” to accommodate the comments of the two Reviewers.

REVIEWERS' COMMENTS:

Reviewer #1 (Remarks to the Author):

I want to thank the authors for the thorough responses to my comments from the previous round of review. I am now mostly happy with the paper, and have just a few more comments after rereading the revised manuscript. The new section on empirical predictions still needs some work, and one aspect of the figures and terminology still confuses me. Otherwise everything seems to be in order, nice work! About the figures (these are interrelated problems – make sure they're all fixed & consistent!): Fig. 3a and 4a: Why are the tangent lines labelled Tangent line 1 and 2 as they are? As far as I can tell, Tangent line 1 belongs to Sex 2 in both figures, so perhaps the labels 1 and 2 should be flipped around? If your intention was to have the steepest of the two tangents be tangent 1, this is the case in 3a but not in 4a. Figure legend, fig. 3a: What do you mean by saying "In this model, tangent line of Sex 1 is greater than that of Sex 2"? Do you mean steeper? If so, change to that, or "the slope of the tangent line of Sex 1 is greater than that of Sex 2." Or do you mean that the tangent line meets the fitness function at a greater investment x ? Figure legend, fig. 4a: Here surely the tangent line of Sex 2 is steeper than that of Sex 1, although you write the opposite. However, the one labelled "tangent line 2" in the figure is steeper than the one labelled "tangent line 1". Please clarify. About the section "Relevance of model assumptions and predictions to empirical research", page 6: Line 30: Grammatical errors: "Similar issues can be rise to the measurement..." change to for example "Similar issues can arise regarding the measurement..." Line 33: "...or even the weather conditions (Öberg et al. 2015)": This seems out of place in what I gather has been a list of different ways parental investment has been measured. This last one is rather a factor that can affect the amount of parental investment given – not a proxy of parental investment itself, unless you are insinuating that the parents are controlling the weather conditions somehow. Rather, the proxy used in Öberg et al. is parental nest visitation rates, and it is these nest visitation rates that are affected by adverse weather. Line 36-39: Whether or not the tangent line of the male's fitness function is steeper than that of the female, surely depends on where (at what investment level x) the 'step' in the function occurs. If males only have a chance to get any fitness at very high parental investments, then the female's fitness function may have the steeper tangent line. But I note that in this paragraph (line 37, 39) you keep talking about which tangent line is 'greater'. Please note the wording here and throughout: the relevant comparison is which tangent line is steeper, or equivalently which tangent line has a greater slope. The phrase 'greater tangent line' is a bit meaningless. Line 36: Grammar error. Change "...then is be expected..." to "...then it is expected..." or "...then it would be expected".

Response to the reviewer

Dear reviewer:

We authors sincerely appreciate the reviewer's careful look at the manuscript to correct the mistakes and clarify the meaning. Due to the endeavor and insight of the reviewer, we were able to fix the glitches that we could not find so far. The followings are the responses to the comments.

Comment 1: Fig. 3a and 4a: Why are the tangent lines labelled Tangent line 1 and 2 as they are? As far as I can tell, Tangent line 1 belongs to Sex 2 in both figures, so perhaps the labels 1 and 2 should be flipped around? If your intention was to have the steepest of the two tangents be tangent 1, this is the case in 3a but not in 4a.

Response 1: It is our mistake to label the graphs as they were in the previous version of the manuscript. We fixed the numbering the of sexes in Fig. 3a and 4a.

Comment 2: Figure legend, fig. 3a: What do you mean by saying "In this model, tangent line of Sex 1 is greater than that of Sex 2"? Do you mean steeper? If so, change to that, or "the slope of the tangent line of Sex 1 is greater than that of Sex 2." Or do you mean that the tangent line meets the fitness function at a greater investment x ?

Response 2: We clarified the meaning of the sentence. The modified sentence in the figure label is "In this model, tangent line of Sex 1 is steeper than that of the Sex 2."

Comment 3: Figure legend, fig. 4a: Here surely the tangent line of Sex 2 is steeper than that of Sex 1, although you write the opposite. However, the one labelled "tangent line 2" in the figure is steeper than the one labelled "tangent line 1". Please clarify.

Response 3: The error was caused by a labeling mistake in Fig 4a. As the labels of Sex 1 and Sex 2 are corrected in Fig. 4a, the sentence mentioned in Comment 3 is now correct.

Comment 4: Line 30: Grammatical errors: "Similar issues can be rise to the measurement..." change to for example "Similar issues can arise regarding the measurement..."

Response 4: We fixed the grammatical errors as pointed by the reviewer.

Comment 5: Line 33: "...or even the weather conditions (Öberg et al. 2015)": This seems out of place in what I gather has been a list of different ways parental investment has been measured. This last one is rather a factor that can affect the amount of parental investment given – not a proxy of parental investment itself, unless you are insinuating that the parents are controlling the weather

conditions somehow. Rather, the proxy used in Öberg et al. is parental nest visitation rates, and it is these nest visitation rates that are affected by adverse weather.

Response 5: We agree to the comment of the reviewer. We removed that part from the sentence.

Comment 6: Line 36-39: Whether or not the tangent line of the male's fitness function is steeper than that of the female, surely depends on where (at what investment level x) the 'step' in the function occurs. If males only have a chance to get any fitness at very high parental investments, then the female's fitness function may have the steeper tangent line.

Response 6: We added the condition that the 'step' should occur at low parental investment for male fitness function to have a greater slope. Please take a look at pg. 6, line 36 to pg. 7, line 5.

Comment 7: But I note that in this paragraph (line 37, 39) you keep talking about which tangent line is 'greater'. Please note the wording here and throughout: the relevant comparison is which tangent line is steeper, or equivalently which tangent line has a greater *slope*. The phrase 'greater tangent line' is a bit meaningless.

Response 7: We changed the wording not to confuse the author throughout the manuscript as indicated by the reviewer.

Comment 8: Line 36: Grammar error. Change "...then is be expected..." to "...then it is expected..." or "...then it would be expected".

Response 8: We fixed the grammatical error as pointed out by the reviewer.